# Transcriptional Inhibition of AGPAT2 Induces Abnormal Lipid Metabolism and Oxidative Stress in the Liver of Nile Tilapia *Oreochromis niloticus*

**DOI:** 10.3390/antiox12030700

**Published:** 2023-03-12

**Authors:** Tiantian Feng, Yifan Tao, Yue Yan, Siqi Lu, Yan Li, Xing Zhang, Jun Qiang

**Affiliations:** 1School of Food Science and Pharmaceutical Engineering, Nanjing Normal University, Nanjing 210023, China; 2Key Laboratory of Freshwater Fisheries and Germplasm Resources Utilization, Freshwater Fisheries Research Center, Ministry of Agriculture and Rural Affairs, Chinese Academy of Fishery Sciences, Wuxi 214081, China; 3Wuxi Fisheries College, Nanjing Agricultural University, Wuxi 214081, China

**Keywords:** *Oreochromis niloticus*, antisense RNA, AGPAT2, transcription suppression, regulatory mechanism

## Abstract

The enzyme 1-acylglycerol-3-phosphate *O*-acyltransferase 2 (AGPAT2) is an intermediate enzyme in triglyceride synthesis. The aim was to study the regulatory mechanism of AGPAT2 on Nile tilapia, *Oreochromis niloticus*. In this study, antisense RNA technology was used to knock-down AGPAT2 in Nile tilapia. Compared with the control groups (transfected with ultrapure water or the blank expression vector), the AGPAT2 knock-down group showed a significantly higher weight gain rate, special growth rate, visceral somatic index, and hepatopancreas somatic index; and significantly increased the total cholesterol, triglycerides, glucose, low-density lipoprotein cholesterol, and insulin levels in serum. In addition, the contents of total cholesterol and triglycerides and the abundance of superoxide dismutase, catalase, and glutathione peroxidase in the liver significantly increased, while the malondialdehyde content significantly decreased. The liver cells became severely vacuolated and accumulated lipids in the AGPAT2 knock-down group. Comparative transcriptome analyses (AGPAT2 knock-down vs. control group) revealed 1789 differentially expressed genes (DEGs), including 472 upregulated genes and 1313 downregulated genes in the AGPAT2 knock-down group. Functional analysis showed that the main pathway of differentially expressed genes enrichment was lipid metabolism and oxidative stress, such as steroid biosynthesis, unsaturated fatty acid biosynthesis, the PPAR signaling pathway, and the P53 pathway. We used qRT-PCR to verify the mRNA expression changes of 13 downstream differential genes in related signaling pathways. These findings demonstrate that knock-down of AGPAT2 in tilapia leads to abnormal lipid metabolism and oxidative stress.

## 1. Introduction

The Nile tilapia (*Oreochromis niloticus*) is an economically important fish, which is farmed worldwide. Its advantages include its diverse diet, fast growth, strong resistance, and the short time needed to reach sexual maturity. With the recent increase in large-scale and intensive aquaculture, there have been frequent occurrences of fish liver diseases, such as hepatobiliary syndrome, chemical liver injury, and fatty liver disease, and these adverse health conditions have restricted the development of aquaculture [1]. Among them, liver diseases characterized by fatty liver or fatty liver injury have attracted much attention from researchers. Fatty liver disease in fish is associated with lipid metabolism disorders, oxidative stress, inflammation, and necrosis of liver cells [2]. In aquaculture, fatty liver disease can be caused by overnutrition, environmental stresses, pathogen infection, and gene mutations [3]. The harmful effects of fatty liver disease include reduced growth and feed utilization, decreased immunity, sensitivity to stresses, and poor product quality, all of which negatively affect the aquaculture industry and market consumers.

The enzyme 1-acylglycerol-3-phosphate *O*-acyltransferase 2 (AGPAT2), also known as lysophosphatidylate-acyltransferase, is an intermediate enzyme in the glycerol-3-phosphate pathway that leads to the biosynthesis of phospholipids and triglycerides (TG) [4,5]. There are five subtypes of AGPAT, among which AGPAT2 is particularly prominent. Its functional inactivation can lead to congenital systemic lipid metabolic disorder type 1, with early manifestations of insulin resistance, diabetes, hypertriglyceridemia, and fatty liver [6]. AGPAT2 is the only subtype of AGPAT with a loss-of-function mutation that results in severe congenital systemic lipodystrophy. AGPAT2 consists of 278 amino acids and contains two highly conserved domains: NHXXXXD and EGTR. It is a member of the acyltransferase family and localizes in the endoplasmic reticulum [7]. AGPAT2 catalyzes the acylation of lysophosphatidic acid to form phosphatidic acid (PA), a key intermediate that can be dephosphorylated by PA phosphatase to form diacylglycerol (DAG) or converted into cytidine diphosphate (CDP)-DAG by CDP-DAG synthases 1 and 2 (CDS1 and 2) for the final synthesis of TG [8]. Accumulation of PA was observed in AGPAT2 knock-out adipocytes, as well as in the livers of AGPAT2 knock-out mice [9,10]. Studies have shown that decreased CDS activity in AGPAT2-deficient liver cells leads to the accumulation of PA in the endoplasmic reticulum, which may be the cause of the increased TG level [11]. Cortes et al. [10] showed that the alternative monoacylglycerol (MAG) pathway for TG synthesis was activated in the liver of AGPAT2^-/-^ mice, leading to elevated TG levels in the liver, and, ultimately, hepatic steatosis. Studies have confirmed that the inhibition of AGPAT2 can activate cell apoptosis and cause oxidative stress [12], and lysophosphatidylate can promote the proliferation of preadipocytes and inhibit their differentiation [13]. In addition, a mutation of AGPAT2 can affect the signal transduction of the peroxisome proliferator-activated receptor (PPAR)-γ pathway, which is strongly related to adipose differentiation, resulting in adipose maturation disorder and subcutaneous adipose atrophy. AGPAT2^-/-^ mice reproduce all the features of human lipodystrophy, including hyperinsulinemia, diabetes, hypertriglyceridemia, and hepatic steatosis [10]. The accumulation of saturated fatty acids is hepatotoxic because they induce hepatocyte apoptosis, whereas unsaturated fatty acids are relatively non-toxic, possibly because they can inhibit the liver lipotoxicity induced by saturated fatty acids by transferring toxic fatty acids into TG storage [14,15]. Most studies on AGPAT2 have focused on diseases in mammals. There are few reports on its mechanism of action and regulation on lipid metabolism in aquatic animals.

Antisense RNAs are a class of RNA molecules whose nucleotide sequences are complementary to, and hybridize with, target RNAs (mainly mRNAs) to produce double-stranded RNAs. This affects the normal processing, modification, and translation of the target RNAs, thereby regulating gene expression [16]. Antisense RNAs, by precisely complementing a particular mRNA, specifically blocks the DNA molecule it is translating, resulting in low or no expression of the target gene [17]. Since Tomizawa et al. [18] first used antisense RNA technology to inhibit replication of the *Escherichia coli* Col E1 plasmid, this technology has been further developed to block or inhibit the expression of specific genes. The use of this technology has resulted in remarkable achievements in the fields of medicine [19], crop development [20], and microbial fermentation [21], but it is seldom used in in vivo models.

Instead of microinjection transfection, our laboratory has introduced specific antisense RNA sequences into eggs through micropores to knock-down *sf1* and *amh* in Nile tilapia. Our previous studies have shown that *sf1* knock-down results in weight gain and gonadal dysplasia in male and female Nile tilapia [16], and that *amh* knock-down results in decreased follicle-stimulating hormone levels and increased follicular atresia in female Nile tilapia [22], and weight gain and gonadal development inhibition in male Nile tilapia [23]. This method is relatively simple, which not only greatly improves the transfection efficiency, but also minimizes the damage to the ovum. In addition, the phenotype of the offspring is stable, so the method is conducive to large-scale production. In this study, tilapia with knocked-down AGPAT2 were successfully created using antisense RNA technology. The changes in the liver characteristics and histology of Nile tilapia were evaluated. The downregulation of AGPAT2 transcripts and AGPAT2 protein were confirmed by qRT-PCR and Western blotting analyses, respectively. Transcriptome analyses showed that the knock-down of AGPAT2 affected the expression of other genes, and their functions and pathways were analyzed. Our results provide new information on the regulatory mechanisms of AGPAT2.

## 2. Materials and Methods

### 2.1. Construction of AGPAT2 Knock-Down Model

#### 2.1.1. Experimental Fish

Nile tilapia from the FFRC were used in this experiment. The fish were kept in a 450 L indoor circulating water tank (temperature 28 °C ± 0.5 °C, pH 7.5 ± 0.2, dissolved oxygen ≥ 6.0 mg/L). The fish were fed with commercial feed at a rate of 5% of their body weight twice per day (8:00 and 16:00). The feed consisted of 32% crude protein and 6% crude fat.

#### 2.1.2. Design of Antisense RNA Sequences

Two antisense RNA sequences were designed for the inhibition of AGPAT2 expression:

Antisense RNA sequence I (anti-AGPAT2-I): 

GCGGTATCATCCACAGCACATCCATAGCGAAACAGCCGCAGAAACTAGCAGCAGCACTGACTCACGCGGGGTGTTTCTGGATGAGGAAGCCGCTAACGGTTT.

Antisense RNA sequence II (anti-AGPAT2-II): 

TAGCGCGGGCTAGCTGTCAGACGGCGCCGCGAGCGACATTTCTGCAAAACTCACCTCATGTTTTCAATGTCTCTGCCTCCGCTCTTTAGTATGCACAGAGGTATAGC.

Antisense RNA encoded DNA fragments were sent to Genewiz Biotechnology Co., Ltd. (Suzhou, China) for synthesis.

#### 2.1.3. PCR Amplification

The two antisense RNA sequences were synthesized, and each cloned into the pcDNA3. 1 expression vector containing the strong CMV promoter (Invitrogen; Thermo Fisher Scientific, Waltham, MA, USA). The clones were used as templates for subsequent PCR amplification. A pair of specific primers was designed for amplification: polyAF1 (GCAGGCACAAGTACCCATCA) and polyAR1 (CAGCACGCATCAGGTCAAAG). The reaction procedure follows the method of Qiang et al. [22].

#### 2.1.4. Preparation of Transfection Reagent

The antisense RNA targeting AGPAT2 (the AGPAT2 knock-down group), the blank expression vector amplification product (negative control group) or the ultrapure water (control group) were used for transfection. Each product was mixed with the transfection agent at a ratio of 1:5 and equilibrated at room temperature for 30 min.

#### 2.1.5. Artificial Insemination and Incubation

A mature female tilapia was selected, and the eggs were obtained by gently squeezing the abdomen. The eggs were placed in three clean, dry stainless steel containers (each containing 200–250 eggs). A buffer solution was prepared according to the method of Qiang et al. [22]. A 1.5 mL aliquot of this buffer was added to each container of eggs to keep the fertilization hole open, and then 0.8 mL of the transfection reagent was added. To allow the antisense RNA carrier to enter the egg through the fertilization hole, each container was gently shaken for 15 min. The negative control group (0.8 mL transfection reagent containing the blank vector) and the control group (0.8 mL transfection reagent containing the ultrapure water) were treated in the same way.

A male fish with well-developed gonads (red and prominent reproductive papillae) was selected, and then 0.2 mL of semen was collected using a disposable dropper. The semen was added to each of the containers. The goose feathers were stirred for 30 s, and then 2 mL of the incubation water was added to complete fertilization.

The fertilized eggs were placed in incubators with a temperature of 28 °C and a water rate of 5.5 L/min to ensure adequate rolling. At 96 h after hatching, the newly hatched larvae were collected and counted. The hatching rate was about 85%. Newly hatched fry (50–60 tails) were placed in a 30 L tank equipped with a recycling water system. The fry were fed four times a day (45.0% crude protein, 8.0% fat) at a rate of 15–20% of their body weight for 30 days.

#### 2.1.6. Management of Experimental Fish

We selected 90 male fish with an average weight of 5.34 ± 0.28 g from the control group, the negative control group, and the AGPAT2 knock-down group. The fish were raised in nine tanks (three tanks per treatment) equipped with inflatable circulation systems. Each 1.2 m^3^ tank contained 30 fish, and each group had three replicates. The fish were fed with commercial feed (32% crude protein, 6% crude fat) at a rate of 5% of their body weight twice a day (8:00 and 16:00). One-third of the water was replaced every 3 days. The tanks had a circulating water system and were continuously aerated to maintain dissolved oxygen of ≥6 mg/L. The pH was 7.5 ± 0.2, and the water temperature was 28 °C ± 0.5 °C. The whole experimental period was 90 days.

#### 2.1.7. Detection of Positive Transfection Efficiency

After feeding to 60 days of age, 12 Nile tilapia were randomly selected from the AGPAT2 knock-down group, the negative control group, and the control group. The fish were deeply anesthetized with 200 mg/L of MS-222 solution and their liver tissues were dissected. Genomic DNA was extracted using the MiniBEST Universal Genomic DNA Extraction Kit Ver 5.0 (Takara Bio Inc., Shiga, Japan). The PCR reaction mixtures contained 20 µL (including 0.5 µL of upstream and downstream primers (F1, TTTTGCGCTGCTTCGCGATGTAC: R1, TCCCAATCCTCCCCCTTGCTG, Concentration 10 mmol/µL), 1 µL genomic DNA, 10 µL Premix Taq (LA Taq 2.0), and 8 µL RNase-free water. The reaction procedure is referred to by Qiang et al. [22].

### 2.2. Sampling

Food was withheld from the fish for 24 h before sampling. Twelve tilapias were selected from each of the AGPAT2 knock-down group, the control group, and the negative control group. After the MS-222 was anesthetized (200 mg/L), a sterilized disposable radiator was used to take blood intravenously from the experimental fish tails. The blood sample was placed in a 4 °C refrigerator, 5000× *g*, and centrifuged for 15 min for the preparation of the serum and stored at −20 °C until analysis. The fish bodies and livers were dissected, weighed, and photographed. In addition, nine fish were selected from each group and their livers were removed after deep anesthesia. The livers were divided into six parts: four parts were stored in cryogenic storage tubes and frozen in liquid nitrogen, and the other two parts were fixed in a 4% *v*/*v* paraformaldehyde solution for tissue structure analysis.

### 2.3. Measurement of Indexes

#### 2.3.1. Calculation of Growth Indexes

Various indexes were calculated as follows:Weight gain rate (WGR) = [final weight (g) − initial weight (g)]/initial weight;(1)
Special growth rate (SGR) = [in final weight(g) − in initial weight (g)]/days of breeding;(2)
Visceral somatic index (VSI) = [visceral mass (g)/body mass (g)] × 100;(3)
Hepatopancreas somatic index (HSI) = [liver mass (g)/body mass (g)] × 100.(4)

#### 2.3.2. Serum Index Determination

A fully automated biochemical analyzer (bs-400; Mindray, Shenzhen, China) was used to detect the contents of triglyceride (TG), total cholesterol (TC), glucose (Glu), insulin (INS), high-density lipoprotein cholesterol (HDL-C) and low-density lipoprotein cholesterol (LDL-C) in serum. The reagents and kits were purchased from the Nanjing Jiancheng Institute of Bioengineering (Nanjing, China).

#### 2.3.3. Determination of Liver Lipid Indexes

Determination of the liver lipid indexes was carried out according to the method proposed by Folch et al. [24]. Briefly, each liver sample (about 0.1 g) was placed in a pre-weighed 1.5 mL tube containing 2–3 grinding beads. Then, nine volumes of a chloroform/methanol mixture were added. The mixture was ground using a high-throughput tissue grinder (60 Hz, 200 s) and then centrifuged for 20 min (4 °C, 3000× *g*). The supernatant was transferred to a new tube and stored at −20 °C until analysis. The TG and TC concentrations in the liver were measured using kits purchased from the Nanjing Jiancheng Institute of Bioengineering.

### 2.4. Determination of Liver Antioxidant Indexes

The liver samples (about 0.1 g) were placed into a 1.5 mL sterile tube containing two quartz grinding beads, and nine volumes of pre-cooled phosphate buffer (PBS, 50 mM, pH = 7.4) were added. The samples were crushed using a high-throughput tissue grinder for 200 s (60 Hz) and then the mixture was centrifuged for 20 min (4 °C, 3000× *g*). The supernatant was liver homogenate. The activity of superoxide dismutase (SOD), catalase (CAT), glutathione peroxidase (GSH-Px) and the content of malondialdehyde (MDA) in the liver were determined with the kit purchased by the Nanjing Jiancheng Institute of Bioengineering

### 2.5. Histological Analyses

According to Tao et al. [25], liver tissue samples were taken, rinsed with saline, and divided into two groups. The first section was fixed in 4% paraformaldehyde for 24 h, dehydrated in graded ethanol and embedded in paraffin wax, which was then cut into 4-μm using a paraffin cutter (Leica RM2235; Leica Microsystems, Wetzlar, Germany). Sections were stained with hematoxylin-eosin (HE) and observed under a light microscope (E100; Nikon, Tokyo, Japan). The second section was quickly frozen in liquid nitrogen and then cut into 10-µm frozen slices using a frozen slicer (Leica 3050S). Sections were stained with oil red O solution for 15 min, separated using 60% isopropyl alcohol for 10 s, rinsed with distilled water for 30 s, re-stained with hematoxylin for 3 min, washed with water for 30 s, and observed under a light microscope. The reagents used for histological staining were purchased from the Nanjing Jiancheng Institute of Bioengineering.

### 2.6. RNA Extraction and Reverse Transcription

Total RNA was extracted from the tilapia liver tissues using the Trizol method, and then quality control was performed using a NanoDrop ND-1000 instrument (NanoDrop, Wilmington, DE, USA). The RNA integrity was then measured using a Bioanalyzer 2100 instrument (Agilent, Palo Alto, CA, USA) and by electrophoresis on agarose gels (0.3 g agarose + 30 mL 1 × TAE) at 180 V. The cDNA was prepared by reverse transcription, according to the instructions of the Prime Script™ II 1st Stand cDNA Synthesis Kit (Takara, Dalian, China). The reverse transcription reaction conditions were 42 °C for 2 min, 37 °C for 15 min, and 85 °C for 5 s.

### 2.7. Library Construction and Transcriptome Sequencing and Analysis

#### 2.7.1. Library Construction

Three fish samples from the AGPAT2 knock-down group and control group were combined to construct each sequencing library. This was repeated three times for each group. To establish an independent sequence library, the Nile tilapia AGPAT2 knock-down group and the control group were analyzed. We constructed three control (CON 1–3) and three AGPAT2 knock-down (AT2 1–3) libraries. The libraries were sequenced using an Illumina Novaseq™ 6000 instrument (LC-Bio Technology Co., Ltd., Hangzhou, China) in PE150 sequencing mode, following standard protocols.

#### 2.7.2. Assembly and Annotation of Transcripts

The raw read data were in fastq format. We used cutadapt (https://cutadapt.readthedocs.io/en/stable/, v1.9, accessed on 5 September 2021) to remove adapter sequences, low-quality sequences, and repeated sequences. The resulting clean data were in fastq.gz format. The clean data were compared with the reference genome using HISAT2 (https://daehwankimlab.github.io/hisat2/, v2.0.4, accessed on 5 September 2021). The initial assembly of the gene or transcript was performed using StringTie software (http://ccb.jhu.edu/software/stringtie/, v1.3.4d, accessed on 5 September 2021). The initial assembly results from all samples were combined, and the final assembly was obtained by merging the transcriptomes from all samples using the gffcompare software (http://ccb.jhu.edu/software/stringtie/gffcompare.shtml, v0.9.8., accessed on 5 September 2021). BLAST was used to compare the valid data with the reference genomes.

#### 2.7.3. Identification of Differentially Expressed Genes

We used reads per kilobase per million reads (RPKM) values to measure the abundance of gene transcripts [26]. The DESeq2 R package (http://www.bioconductor.org/packages/release/bioc/html/DESeq2.html accessed on 5 September 2021) was used to detect significantly differentially expressed genes (DEGs) between the control group and the AGPAT2 knock-down group, with the criteria |log2 fold change| ≥ 1 and *p* < 0.05 [27]. All the DEGs were subjected to Gene Ontology (GO) and Kyoto Encyclopedia of Genes and Genomes (KEGG) analyses to determine their potential functions and related metabolic pathways.

### 2.8. qRT-PCR Analyses

Real-time fluorescence quantitative PCR was performed using the SYBR^®^ Premix Ex Taq^TM^ II kit (Takara). The transcript levels of each gene were measured using ABI QuantStudio 5 real-time PCR system (Applied Biosystems, Foster City, CA, USA) and gene-specific primers (Table 1). The thermal cycling conditions for the PCR were 95 °C pre-denaturation for 30 s, 95 °C for 3 s, 60 °C for 30 s (40 cycles). To ensure primer specificity, the PCR amplification curve and melting curve were confirmed after each operation, and each sample was analyzed three times. The relative transcript level of each gene was calculated using the 2^−∆∆Ct^ method [28]. The tilapia *β-actin* gene was used as an internal reference.

### 2.9. Western Blot Analysis

Antibody preparation: New Zealand white rabbits were immunized with AGPAT2 or *β*-actin proteins of the Nile tilapia to produce polyclonal antibodies. After the titer of rabbit serum was identified using an ELISA kit (Nanjing Jiancheng Institute of Bioengineering), all sera were collected and purified using affinity column chromatography to select the best rabbit serum for each antigen. Then, the protein was purified by sodium dodecyl sulfate polyacrylamide gel electrophoresis (SDS-PAGE). The polyclonal antibodies against AGPAT2 and *β*-actin were determined by the ELISA kit. The antibodies were synthesized by the Hua’an Biotechnology Co. (Beijing, China), and were specifically prepared and tested in accordance with the company’s standard procedures.

Western blot analyses were conducted as described by Qiang et al. [29]. Briefly, 0.1 g of tilapia liver tissue was cut into pieces, frozen in liquid nitrogen, ground into a powder, and then 1 mL of protein cracking solution (RIPA) was added. After 30 min incubation on ice, the mixture was centrifuged at 4 °C at 12,000 RPM for 15 min. The supernatant was transferred to a new tube and the protein concentration was detected with a protein kit (Nanjing Jiancheng Institute of Bioengineering). An aliquot of the sample was mixed with four volumes of 5× SDS-PAGE protein loading buffer, and then denatured by heating in a water bath at 100 °C. The sample was separated by electrophoresis at 80 V for 15 min and then 100 V for 1 h. The separated proteins were electrophoretically transferred to a membrane (2 h, 220 mA). The membrane was blocked with 10% *w*/*v* skimmed milk solution for 1 h, washed with PBST four times (8 min per wash), then with 5% skimmed milk solution, and then incubated with the primary antibody (dilution ratio of 1:1000) at 4 °C overnight. The next day, the secondary antibody (dilution ratio: 1:5000) was added and the membrane was gently agitated at room temperature for 1 h. The membrane was washed four times with PBST (8 min per wash). The color was developed using the Immobilon Western HRP luminescent reagent (Millipore, Billerica, MA, USA). Images were acquired using the Odyssey two-color infrared fluorescence imaging system (Li-Cor Biosciences, Lincoln, NE, USA). Image J software was used to detect the gray value of each protein band and normalize it to that of *β*-actin.

### 2.10. Statistical Analysis

All results shown in figures and tables (growth performance indexes, serum/liver biochemical indexes, qRT-PCR results) are mean ± standard error (mean ± SE). The data were analyzed using SPSS 26.0 statistical software. The data were first tested for normality and homogeneity of variance using the Shapiro–Wilk test and the Levene’s test, followed by the one-way ANOVA (independent samples *t*-test) for comparisons between groups, followed by Duncan’s multiple comparison post-test. Differences were considered significant at *p* < 0.05.

## 3. Results

### 3.1. Positive Transfection Efficiency

The liver tissues of 60-day-old fish were analyzed by the RT-PCR using specific primers to detect the positive transfection rate. In the agarose gel electrophoresis analysis, fragments of about 1100–1200 bp were detected in the AGPAT2 knock-down group (Figure 1a), fragments of 1000 bp were detected in the negative control group (Figure 1b, B1–B12), while no corresponding fragments were detected in the control group (Figure 1b, B13–B24). The fragments were excised and cloned. Ten monoclonal colonies were sent to Genewiz Biotechnology Co. Ltd. for sequencing. The results confirmed that the two transfected antisense RNA sequences were present in the fish in the AGPAT2 knock-down group.

### 3.2. Effects of AGPAT2 Knock-Down on Growth of Nile Tilapia

To observe the changes caused by the AGPAT2 knock-down, the Nile tilapia were weighed, and various growth indexes were calculated. The VSI and HSI of the Nile tilapia in the AGPAT2 knock-down group were significantly higher, and their WGR and SGR were significantly lower, compared with those in the control group and negative control group (Table 2, *p* < 0.05).

### 3.3. Effect of AGPAT2 Knock-Down on Serum Biochemical Parameters of Nile Tilapia

Compared with the control group and negative control group, the AGPAT2 knock-down group showed significantly increased levels of TC, TG, Glu, LDL-C, and insulin in serum (*p* < 0.05), but no significant change in serum HDL-C levels (*p* > 0.05) (Table 3).

### 3.4. Effect of AGPAT2 Knock-Down on Liver Biochemical Indexes of Nile Tilapia

As shown in Table 4, the levels of TC and TG in the liver were significantly higher in the AGPAT2 knock-down group than in the control group and the negative control group (*p* < 0.05). The MDA content significantly decreased (*p* < 0.05) and the SOD, CAT, and GSH-PX activities significantly increased (*p* < 0.05) in the AGPAT2 knock-down group compared with the control group and the negative control group.

### 3.5. Effect of AGPAT2 Knock-Down on Liver Tissue Structure of Nile Tilapia

Liver slices from the Nile tilapia were stained with HE to reveal the accumulation of lipid vesicles (Figure 2d–i). Compared with those in the control and negative control groups, the Nile tilapia with knocked-down AGPAT2 accumulated more and larger lipid droplets in the liver cells. Oil red O staining was used to detect neutral lipids and lipid droplets in the liver tissues. In the liver of Nile tilapia with knocked-down AGPAT2, the red lipid droplets stained with oil red O occupied almost the entire area of the liver section, while there were only a few red lipid droplets in the liver of those in the control and negative control groups. In addition, the livers of the Nile Tilapia in the control and negative control groups were pink, whereas the livers of those in the AGPAT2 knock-down group were yellow (Figure 2a–c).

### 3.6. Antisense RNA Downregulated AGPAT2 Transcripts and AGPAT2 Protein in Nile Tilapia Liver

To analyze the inhibitory effect of antisense RNA on the target gene and its encoded protein, qRT-PCR and Western blotting analyses were conducted to detect the level of AGPAT2 transcripts and AGPAT2 protein in the liver. The transcript level of AGPAT2 was significantly lower in the AGPAT2 knock-down group than in the control group and negative control group (Figure 3a), and the protein content of AGPAT2 was also significantly lower in the AGPAT2 knock-down group than in the control group and negative control group (Figure 3b).

### 3.7. Transcriptomic Analysis to Determine the Effects of AGPAT2 Knock-Down on Liver Lipid Metabolism

#### 3.7.1. Sequencing of mRNA Libraries and Identification of DEGs

We detected significant DEGs in the livers between the AGPAT2 knock-down group and the control group using sequence data generated with the Illumina Novaseq 6000 platform. The number of clean reads in the libraries ranged from 37,271,364 to 495,433,150. The six libraries (CON 1–3, AT2 1–3) had Q20 values ranging from 99.75% to 99.95%, and GC content ranging from 45% to 47% (Appendix A). These results confirm the reliability of the transcriptome sequencing analysis and the high quality of the sequencing data. A total of 22,283 genes were sequenced, of which 3437 were upregulated and 13,559 genes were downregulated. After applying the criteria for significant DEGs (|log2 fold change| ≥ 1 and *p* < 0.05), we detected 1785 DEGs, including house-keeping genes *ldha*, *nono*, *arhgdia*, and *rpl11* that were significantly down-regulated. There were 472 upregulated genes and 1313 downregulated genes in the AGPAT2 knock-down group compared with the control group (Figure 4).

#### 3.7.2. Functional Annotations by GO and KEGG Analyses

The selected DEGs were subjected to functional classification and enrichment pathway analyses using the tools in the GO and KEGG databases. The DEGs in the AGPAT2 knock-down group (compared with the control group) were significantly enriched in 212 GO terms, including 97 in the biological process category, 33 in the cellular component (CC) category, and 82 in the molecular function (MF) category (Appendix A). The top three significantly enriched BP terms were transmembrane transport, protein folding, and vesicle mediated transport. The top three significantly enriched CC terms were cytoplasm, endoplasmic reticulum membrane, and extracellular space. The top three significantly enriched MF terms were ATP binding, metal ion binding, and binding (Figure 5a). The results of the KEGG analyses showed that the AGPAT2 knock-down mainly affected the lipid metabolism (Figure 5b), including steroid biosynthesis; fructose and mannose metabolism; the PPAR signaling pathway; biosynthesis of unsaturated fatty acids; glycine, serine, and threonine metabolism; amino sugar and nucleotide sugar metabolism; and glycosylphosphatidylinositol (GPI) signaling. We further investigated the KEGG pathways associated with lipid metabolism and oxidative stress, including the PPAR signaling pathway, the biosynthesis of unsaturated fatty acids, and the p53 signaling pathway.

#### 3.7.3. qRT-PCR Validation of Gene Expression Patterns

The KEGG pathways enriched with DEGs between the AGPAT2 knock-down group and the control are shown in Table 5. Thirteen DEGs were related to lipid metabolism and oxidative stress, as illustrated in the heatmap in Figure 6a. To verify the accuracy of the RNA sequence data, we performed qRT-PCR analysis of the 13 DEGs associated with lipid metabolism and oxidative stress. The correlation analysis showed a relatively high linear correlation (R^2^ = 0.8469) for gene transcript levels determined using these two methods, indicating that the transcriptome sequencing data were reliable (Figure 6b). The results of the qRT-PCR analyses showed that, compared with the control group, the AGPAT2 knock-down group showed significantly higher transcript levels of *pisd*, *aspg*, *pnpla2*, and *gadd45β*, and significantly lower transcript levels of *lypla2*, *fabp7a*, *pparαb*, *scd*, *elovl6*, *fasn*, *chkb*, *gnpat*, and *cdk2* in the liver (Figure 6c). The trends in gene expression detected by the qRT-PCR were the same as those detected in the RNA-seq data (Figure 6d).

## 4. Discussion

Fatty liver disease induced by intensive farming has become common in tilapia, and is one of the factors limiting the development of the aquaculture industry [30]. Fatty liver disease leads to hepatocyte steatosis, hepatocyte injury, an inflammatory response, and liver fibrosis, and is closely related to insulin resistance and oxidative stress [31,32].

The antisense RNA technique used in this study effectively knocked-down AGPAT2 resulting in decreased levels of both the gene transcript and the protein. Below, we discuss the changes in the Nile tilapia after AGPAT2 knock-down and explore the possible mechanisms underlying these changes.

### 4.1. AGPAT2 Knock-Down Slows Growth of Nile Tilapia

The growth status of fish is indicated by the WGR and SGR [33], while the health status is indicated by the HSI and VSI to a certain extent [25]. Studies have shown that patients with congenital systemic lipodystrophy (CGL 1) with a mutation in the AGPAT2 gene show a reduction in subcutaneous fat deposition, dysplasia, and other conditions [34]. Cortes et al. [10] found that AGPAT2^(−/)^ mice had symptoms resembling those of CGL1 patients. In this study, we found similar results after AGPAT2 knock-down. In these fish, the yellow color of the liver may have been because AGPAT2 knock-down led to disordered lipid metabolism and an increased TG content in the liver [35], resulting in an increased load and growth inhibition.

### 4.2. AGPAT2 Knock-Down Causes Disordered Lipid Metabolism in Nile Tilapia

Blood is an important transport system that participates in the regulation of lipid metabolism. The TC and TG in serum are mainly synthesized in the liver, so changes in blood lipids can indirectly reflect the status of lipid metabolism in the liver. In general, the changes in TC and TG in serum are similar [36]. In this study, serum TG levels were significantly higher in the AGPAT2 knock-down group than in the control group. Studies have shown that, in AGPAT2 knock-out mice, activation of an alternative MAG pathway for TG biosynthesis leads to hepatic steatosis [10]. When excess TG is transported from the liver, it causes the serum TG level to rise. Studies have shown that HDL can transport cholesterol from the blood and tissues to the liver for metabolism, while LDL has an opposite function, acting as a transporter of cholesterol from the liver to the peripheral tissues [25]. When excess lipids accumulate in the liver, various metabolic pathways are activated, and lipids are transported to peripheral tissues via lipoproteins [37]. Thus, high-fat feeding activates endogenous lipid transport in fish and increases lipid transport from the liver to the peripheral tissues. In this experiment, the LDL-C content in serum was significantly higher in the AGPAT2 knock-down group than in the control group and negative control group, while the HDL-C content in serum did not differ significantly among the three groups. This result indicates that AGPAT2 knock-down affected endogenous lipid transport in the fish body, and increased lipid transport from the liver to the peripheral tissues. This was also consistent with the increased TC and TG contents in serum in the AGPAT2 knock-down group.

Glucose is an important energy source for many biological processes in fish, and its contents in serum can reflect the metabolic status of the body and the functions of cells and organs [38]. Insulin can reduce the blood sugar concentration. An increase in the serum insulin level promotes glucose uptake in the liver, thus contributing to the control of blood sugar levels [25]. We detected increases in the serum glucose and insulin concentrations in the AGPAT2 knock-down group in this study. Elevated blood glucose is indicative of an increased metabolic burden in the liver, suggesting that AGPAT2 knock-down may lead to severe insulin resistance in Nile tilapia, similar to that observed in AGPAT2^(−/−)^ mice [10].

Hepatic lipid deposition often occurs in cultured fish, and is closely related to metabolic disorders [39]. In this experiment, the TC and TG levels in the liver were significantly higher in the AGPAT2 knock-down group than in the control groups. This may be because AGPAT2 knock-down increased the lipid metabolic load in the liver of Nile tilapia, leading to fat accumulation because the lipids could not be utilized as fast as they were produced. In AGPAT2 knock-out mice, activation of an alternative MAG pathway for TG biosynthesis leads to hepatic steatosis [10].In the present study, HE and oil red O staining of liver sections revealed abundant fat droplets in the liver of Nile tilapia in the AGPAT2 knock-down group, as well as abnormal liver tissue structure resulting in liver damage. These results show that AGPAT2 knock-down causes liver fat deposition and liver damage in Nile tilapia.

### 4.3. AGPAT2 Knock-Down Increases the Antioxidant Capacity of Nile Tilapia

Studies have shown that excessive fat accumulation in fish causes oxidative stress, resulting in the generation of reactive oxygen species (ROS). If ROS are not detoxified quickly by the antioxidant system, they can cause serious oxidative damage to cells and organs, thus negatively affecting fish health [40,41]. In this study, we evaluated various biochemical indexes of the Nile tilapia liver to determine the effects of AGPAT2 knock-down on the liver’s antioxidant capacity. The final product of lipid peroxidation is MDA, and an increase in MDA content indirectly affects the degree of cellular damage caused by ROS and free radicals [42]. The reduced MDA content in the liver of the AGPAT2 knock-down group suggests that liver fat deposition may lead to excessive ROS production, but they are effectively scavenged by the antioxidant system. The main antioxidant enzymes are SOD, CAT, and GSH-Px. These enzymes work together to effectively remove and balance excess ROS, thereby playing an important role in maintaining the dynamic balance of ROS in the body [43,44]. When the body is under oxidative stress, there is increased expression of genes encoding SOD and in SOD enzyme activity. Therefore, the transcript levels of SOD genes can reflect the antioxidant capacity of the organism [45]. In this study, the SOD, CAT, and GSH-Px activities in the liver were higher in the AGPAT2 knock-down group than in the control group, indicating that AGPAT2 knock-down leads to liver metabolic disorder, activation of the antioxidant system to remove excess ROS, and an increased antioxidant defense capacity and reduced oxidative stress response in the liver.

### 4.4. AGPAT2 Knock-Down Affects Regulatory Mechanisms of Lipid Metabolism and Oxidative Stress in Nile Tilapia

To explore the potential regulatory mechanisms associated with the observed biochemical and physiological changes in the AGPAT2 knock-down group, we conducted transcriptome analysis. The KEGG pathway enrichment analysis showed that the PPAR signaling pathway and biosynthesis of unsaturated fatty acids were enriched with DEGs between the AGPAT2 knock-down group and the control group. Lipid metabolism and deposition include lipogenesis, β-oxidation, and long chain polyunsaturated fatty acid biosynthesis. Fatty acid β-oxidation is an important process in lipid metabolism in the liver. Incomplete oxidative decomposition of fatty acids in the liver leads to the accumulation of lipids, resulting in a variety of liver diseases. PPARs are ligand-activated nuclear receptors in the nuclear hormone receptor family that control many intracellular metabolic processes [46]. PPARα, one of the PPAR subtypes, is highly expressed in the liver and regulates the balance of glucose and lipid metabolism in the body [47,48]. After activation, PPARα can regulate the uptake, binding, and oxidation of fatty acids, and has a significant effect on improving vascular injury, inflammation, oxidative damage, and liver injury caused by apoptosis of the liver cells, thereby helping to maintain the balance of lipid metabolism [49]. Inhibition of PPARα expression reduces the hepatic mitochondrial fatty acid β-oxidation capacity, triggers lipid accumulation, and affects lipid metabolism. We observed that in AGPAT2 knock-down fish, the hepatic adipocytes were hollow and accumulated large lipid droplets, suggesting that the gene knock-down affected the PPAR metabolic pathway, leading to oxidative stress and hepatic lipid deposition. FABP7 is closely related to lipid metabolism. Its main function is to bind and transport polyunsaturated fatty acids (PUFAs) [50]. FABP7 strongly binds ω-3 PUFA, suggesting that it plays a role in cell differentiation or proliferation by regulating lipid metabolism or signal transduction [51,52]. Fish can produce monounsaturated fatty acids directly by desaturation. Stearoyl-coenzyme A desaturase is a rate-liming enzyme in monounsaturated fatty acid synthesis, converting saturated fatty acids, e.g., stearoyl CoA, to monounsaturated fatty acids (i.e., stearic acid, 18:0) [53,54]. In fish, long-chain polyunsaturated fatty acids (e.g., arachidonic acid, eicosapentaenoic acid, and docosahexaenoic acid) are synthesized from C18 fatty acids (e.g., linoleic acid, 18:2n6; linolenic acid, 18:3n3) via the activities of desaturases (e.g., fatty acid desaturase, Fads) and elongases (e.g., Elovl) [55]. All of these genes regulate lipid metabolism in fish [56]. We detected significantly decreased transcript levels of *pparαb*, *fabp7a*, *scd*, and *elovl6* in tilapia with knocked-down AGPAT2, compared with the control group. This may be because AGPAT2 knock-down affects the metabolism of PPAR binding receptors and unsaturated fatty acids, and the resulting downregulation of *pparαb*, *fabp7a*, *scd*, and *elovl6* leads to the defective β-oxidation of fatty acids and the blockage of the PUFA transport, resulting in fat accumulation in the liver.

The PPAR signaling pathway also affects the expression of key genes in fat metabolism. FASN plays an important role in fatty acid synthesis, and is closely related to fat metabolism [57]. Lysophosphatidylase 2 (LYPLA2) is involved in lipid metabolism, and helps to stabilize the lysophosphatidyl content in the body [58]. GPAT is a rate-limiting enzyme in the lipid anabolic pathway [59]. Mice with knocked-out *gpat4* lost weight when fed with a normal diet, and showed some resistance to obesity even when fed with a high-sugar, high-fat diet [60].We observed a similar phenomenon in AGPAT2 knock-down tilapia, which showed inhibited growth compared with that of the control group when fed with a normal diet. PNPLA2 plays a key role in maintaining lipid metabolic homeostasis by catalyzing the initial step of triglyceride lipolysis in adipocytes [61]. Pnpla2-deficient mice showed an impaired rate of triglyceride hydrolysis in adipocytes and increased adipose tissue mass [62]. CHKB plays a key role in the phospholipid synthesis pathway [63]. Both ASPG [64] and PISD [65] play key roles in adipocyte proliferation, and PISD is located in the inner mitochondrial membrane, which is responsible for phosphatidyl ethanolamine (PE) production and lipid metabolism [65]. In threespine stickleback (*Gasterosteus aculeatus*), AGPAT2 transcription levels have been found to increase in the pectoral muscle in response to cold acclimation and may contribute to mitochondrial phospholipid biosynthesis required for mitochondrial biosynthesis [66]. The expression levels of these key enzymes are important for fat anabolism. In the present study, compared with the control group and the negative control group, the AGPAT2 knock-down group showed significantly lower transcript levels of *fas*, *gpat4*, *chkb*, and *lypla2* in the liver, and significantly increased transcript levels of *pnpla2*, *aspg*, and *pisd*. These findings show that AGPAT2 knock-down causes the abnormal expression of key genes in fat synthesis, resulting in abnormal fat synthesis and the proliferation of fat cells in the body, and ultimately, lipodystrophy.

The high levels of ROS under oxidative stress can exceed the scavenging ability of the antioxidant defense system, leading to damage to mitochondrial DNA and mitochondrial biosynthesis [67]. Moreover, studies have shown that AGPAT2 knockout can affect adipocyte differentiation and mitochondrial morphology in mice [68]. P53 plays a role in metabolism, growth, and development, as well as aging. Studies have shown that p53 overexpression is associated with increased cell proliferation and can activate cell necrosis pathways under oxidative stress [69]. Oxidative stress can induce changes in the p53 signaling pathway at the gene expression level. Cdk2, which is activated by binding to cyclin B1 or phosphorylation by cdk-activated kinase (CAK), is closely associated with the G1 to S phases of cell division, and induces cell division from the G2 phase [70]. However, GADD45A stimulated by p53 can inhibit this process. Recent studies have found that GADD45B is related to the repair of tissues damaged by oxidative stress, ionizing radiation, and infection [71]. In this study, the transcript level of *cdk2* in the liver was significantly decreased in the AGPAT2 knock-down group, while the transcript levels of *gadd45β* were significantly increased. These results suggest that elevated ROS levels in AGPAT2 knock-down tilapia may induce oxidative stress responses that cause cells to stall in the G1 phase instead of entering the division phase. This may lead to apoptosis, the specific mechanism of which will be further studied in the subsequent Nile tilapia.

## 5. Conclusions

We report the application of antisense RNA technology to create an animal gene defect model, i.e., a Nile tilapia liver response model based on AGPAT2 knock-down (Figure 7). During the early development of Nile tilapia, the transcription of AGPAT2 is inhibited, so the transcript levels of AGPAT2 and the abundance of AGPAT2 protein are significantly downregulated. This leads to inhibition of growth and development, increases in serum lipid metabolism indexes and blood glucose levels, and ultimately, insulin resistance. The liver accumulates lipids, its cells become vacuolated, and the activity of liver antioxidant enzymes increases. The knock-down of AGPAT2 leads to abnormalities in the PPAR signaling pathway, biosynthesis of unsaturated fatty acids, and the p53 signaling pathway, leading to oxidative stress and changes in the regulation of lipid metabolism.

## Figures and Tables

**Figure 1 antioxidants-12-00700-f001:**
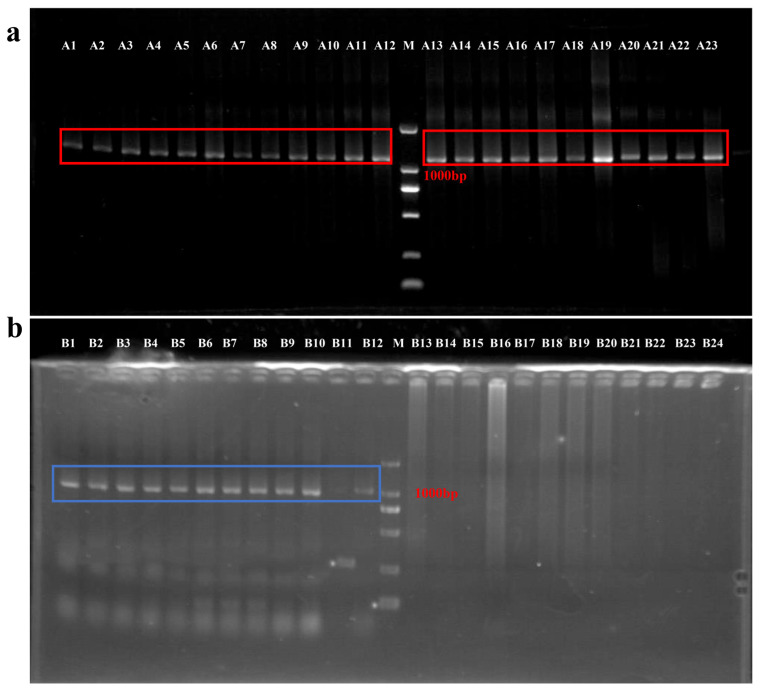
Detection of positive transfection rate in Nile tilapia in the analysis of liver tissue. M. marker. (**a**) PCR analysis of liver tissue from AGPAT2 knock-down group transfected with antisense RNA fragments. A distinct band at ~1100 bp (plasmid 1000 bp + antisense RNA fragment of ~100 bp) is visible. A1–A23 (red boxes) indicate 23 replicates from the AGPAT2 knock-down group. (**b**) B1–B12 (blue boxes) represent negative controls (*n* = 12 replicates) from which a 1000-bp band was amplified (plasmid 1000 bp). B13–B24 represent controls (*n* = 12 replicates) with no obvious band at 1000–1100 bp.

**Figure 2 antioxidants-12-00700-f002:**
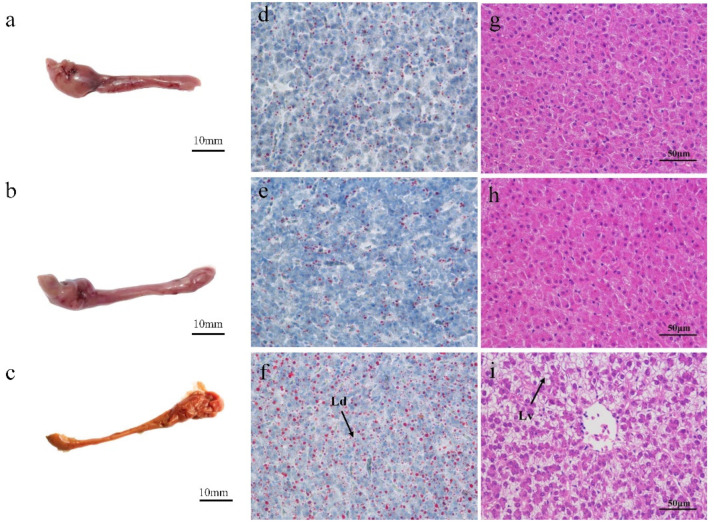
Effect of AGPAT2 knock-down on growth and structure of the liver in Nile tilapia. (**a**–**c**) Nile tilapia livers; (**d**–**f**) oil red O staining of liver tissues of Nile tilapia. Magnification 400×; (**g**–**i**) hematoxylin-eosin staining of liver tissues of Nile tilapia. (**a**,**d**,**g**) Control group; (**b**,**e**,**h**) negative control group; (**c**,**f**,**i**) AGPAT2 knock-down group. Ld: lipid droplet; Lv: lipid vacuole.

**Figure 3 antioxidants-12-00700-f003:**
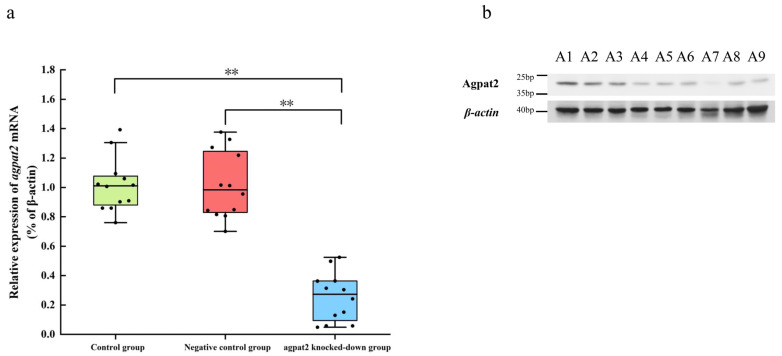
Downregulation of AGPAT2 transcript levels and AGPAT2 protein levels in Nile tilapia liver tissue by antisense RNA technology. (**a**) Transcript levels of AGPAT2 in liver tissues of the AGPAT2 knock-down, control, and negative control groups as determined by qRT-PCR (mean ± SE, *n* = 12 replicates per group). ** indicates significant difference (*p* < 0.05); (**b**) representative images of Western blots showing an abundance of AGPAT2 protein in liver tissues of the control and AGPAT2 knock-down groups: A1–A2, control; A3, negative control; A4–A9, AGPAT2 knock-down. The *β-actin*/β-actin was used as the reference gene transcript/protein.

**Figure 4 antioxidants-12-00700-f004:**
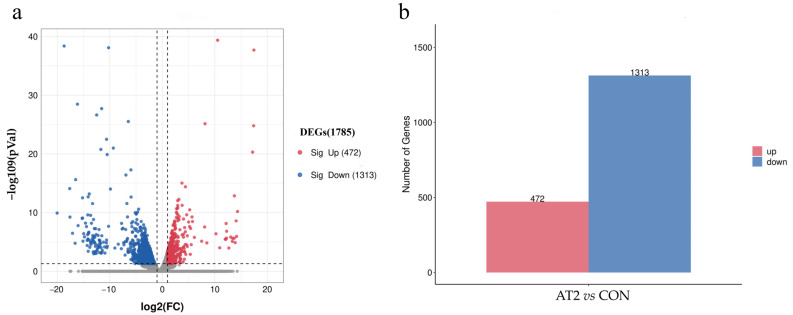
(**a**) Volcano plot of differentially expressed genes (DEGs) between the control group and AGPAT2 knock-down group. (**b**) Histogram of differentially expressed genes (DEGs) between the control group and AGPAT2 knock-down group.

**Figure 5 antioxidants-12-00700-f005:**
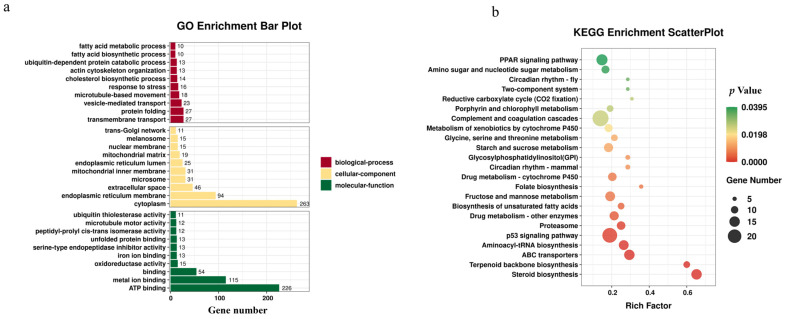
Functional and pathway annotations of DEGs between CON (control group) and AT2 (AGPAT2 -knock-down group). (**a**) Gene Ontology terms significantly enriched with DEGs in the molecular function, cellular component, and biological process categories; (**b**) top 23 KEGG pathways significantly enriched with DEGs.

**Figure 6 antioxidants-12-00700-f006:**
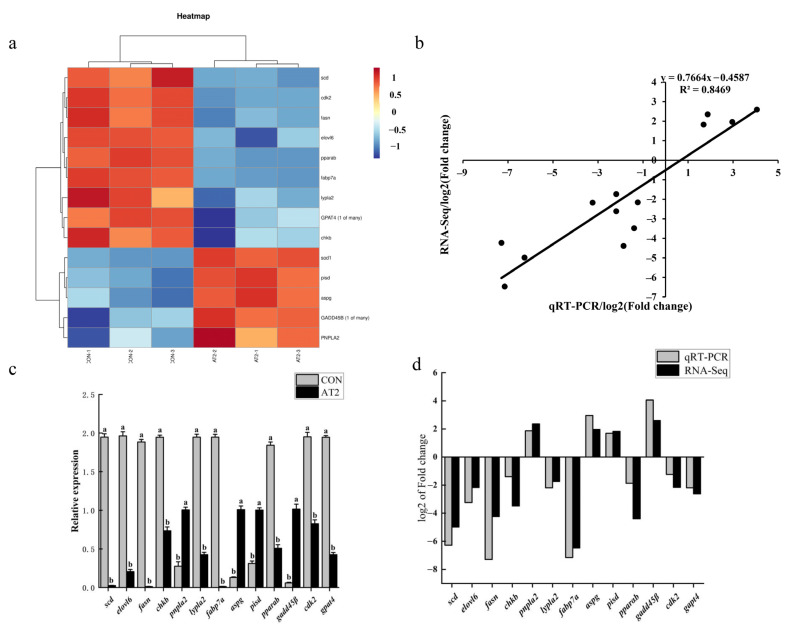
Overview of important DEGs in Nile tilapia. (**a**) Heat map of 14 selected DEGs in CON (control group) and AT2 (AGPAT2 knock-down group) (criteria for identification of DEGs: |log2 fold change| ≥ 1 and *p* < 0.05). Highly expressed genes are shown in red; genes expressed at low levels are shown in dark blue; (**b**) correlation analysis between RNA-Seq and qRT-PCR validation results of 13 genes; (**c**) transcript levels of DEGs (*n* = 9 replicates) in CON (control group) and AT2 (AGPAT2 knock-down group); (**d**) comparison of transcriptome data and qRT-PCR data. Different letters indicate significant differences (*p* < 0.05) between the two groups.

**Figure 7 antioxidants-12-00700-f007:**
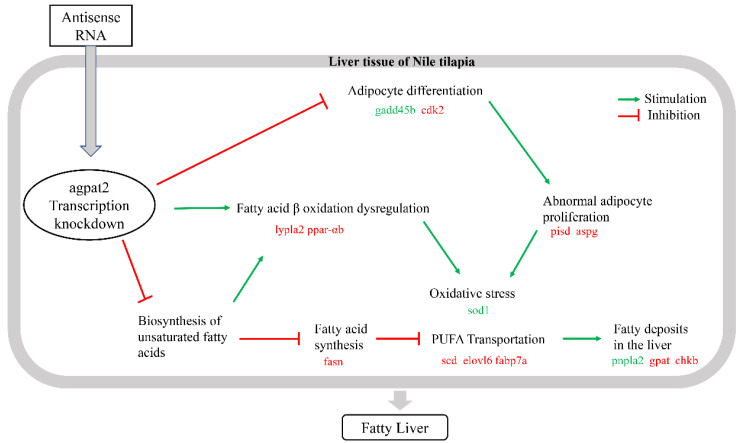
Schematic diagram showing the effects of AGPAT2 knock-down on the regulation of lipid metabolism and oxidative stress in the liver of Nile tilapia.

**Table 1 antioxidants-12-00700-t001:** Specific primer sequences used for qPCR analyses in this study.

Gene	Primer Sequence (5′-3′)	GenBank Number
*agpat2*	F:GCAGGCACAAGTACCCATCA	XM_005454514.4
	R:CAGCACGCATCAGGTCAAAG	
*scd*	F:ACAAGCTCTCCGTGCTGGTCAT	XM_005471382.2
	R:GCAGAGTTGGGACGAAGTAGGC	
*elovl6*	F:ACAGTTCAACGAGGACGAAGC	XM_003443399.5
	R:AGCAAGGGTGAGTGACCACAG	
*fasn*	F:CCAGAATCAGCCTGTGGAGTA	XM_003454056.5
	R:GTTTCAGCCTCAGACTCGTTG	
*chkb*	F:TGTGGAAGATGAACCTCGCC	XM_003440553.5
	R:CAGGGGGAAGACGCCATAAA	
*pnpla2*	F:GCCTCATTCGACGAGCAGAT	XM_003440346.5
	R:TCCAGGGTGAGAGTGTAGGG	
*lypla2*	F:TCGTGAACTGGGACAGCAAAG	XM_003458537.4
	R:CCGTCAAAGTATCCGCCCAC	
*fabp7a*	F:TGCAGAAGTGGGATGGCAAA	XM_003442929.5
	R:GGCTTTCTCATACGTGCGGA	
*aspg*	F:AGAAAGACCGGAGCCCATTT	XM_005476978.4
	R:TGGATCACTCCGCTCACTTG	
*pisd*	F:CTGCCCTTCCTATTGGTGACT	XM_013276568.3
	R:GACTCACGCCACCCTTGA	
*pparαb*	F:AACCATCTCCTTCTGAGCGG	XM_019361775.2
	R:CACTGGACAGACACCAAAGC	
*gadd45β*	F:CTGCTCAGAAACAAGGCTGC	XM_003442905.5
	R:CCCAGAGACTCGCAAGATGG	
*cdk2*	F: CCCCGGTGACTCGGAAATAG	XM_003441494.5
	R: GATAAATCCTGCCGAGCCCA	
*gpat4*	F: AGGCATCTGTGTCGCTAACC	XM_003452655.2
	R: GAATGACCCCCATCAAGCCA	
*β-actin*	F:CCACACAGTGCCCATCTACGA	EU887951.1
	R:CCACGCTCTGTCAGGATCTTCA	

**Table 2 antioxidants-12-00700-t002:** Effects of AGPAT2 knock-down on growth indexes.

	Control Group (*n* = 12)	Negative Control Group (*n* = 12)	AGPAT2 Knock-Down Group (*n* = 12)
Initial body weight (g)	2.99 ± 0.03	3.01 ± 0.02	3.01 ± 0.04
Final body weight (g)	144.91 ± 4.86 ^b^	148.44 ± 3.56 ^b^	107.28 ± 12.54 ^a^
WGR (%)	47.42 ± 1.63 ^b^	48.33 ± 1.23 ^b^	38.55 ± 3.02 ^a^
SGR (%)	4.31 ± 0.12 ^b^	4.33 ± 0.09 ^b^	4.07 ± 0.08 ^a^
VSI (%)	9.52 ± 0.36 ^b^	9.25 ± 0.38 ^b^	12.41 ± 0.56 ^a^
HSI (%)	0.86 ± 0.04 ^b^	0.84 ± 0.04 ^b^	1.30 ± 0.02 ^a^

Note: In each row, different superscript letters indicate significant difference (*p* < 0.05).

**Table 3 antioxidants-12-00700-t003:** Serum indexes for control, negative control, and AGPAT2 knock-down groups.

	Control Group	Negative Control Group	AGPAT2 Knock-Down Group
TC (nmol/g)	1.84 ± 0.07 ^b^	1.97 ± 0.04 ^b^	3.10 ± 0.21 ^a^
TG (nmol/g)	0.23 ± 0.01 ^b^	0.30 ± 0.01 ^b^	0.65 ± 0.11 ^a^
Glu (nmol/g)	4.58 ± 0.26 ^b^	4.44 ± 0.03 ^b^	6.40 ± 0.54 ^a^
HDL-C (mmol/L)	1.13 ± 0.04	1.13 ± 0.00	1.04 ± 0.03
LDL-C (mmol/L)	0.35 ± 0.01 ^b^	0.35 ± 0.00 ^b^	0.64 ± 0.02 ^a^
INS (mmol/L)	13.01 ± 0.48 ^b^	13.00 ± 0.05 ^b^	18.13 ± 0.86 ^a^

Note: In each row, different superscript letters indicate significant difference (*p* < 0.05).

**Table 4 antioxidants-12-00700-t004:** Liver indexes for control, negative control, and AGPAT2 knock-down groups.

	Control Group	Negative Control Group	AGPAT2 Knock-Down Group
TC (nmol/g liver)	1.02 ± 0.06 ^b^	1.07 ± 0.06 ^b^	2.58 ± 0.12 ^a^
TG (nmol/g liver)	4.41 ± 0.23 ^b^	4.52 ± 0.39 ^b^	8.05 ± 0.29 ^a^
SOD (U/mg prot)	7.71 ± 0.38 ^b^	7.73 ± 0.40 ^b^	9.75 ± 0.41 ^a^
CAT (U/mg prot)	43.83 ± 1.25 ^b^	42.57 ± 1.11 ^b^	64.53 ± 2.50 ^a^
MDA (mmol/mg prot)	24.03 ± 1.28 ^b^	25.05 ± 1.79 ^b^	14.63 ± 1.07 ^a^
GSH-PX (U/mg prot)	156.99 ± 11.59 ^b^	159.36 ± 4.30 ^b^	338.26 ± 13.67 ^a^

Note: In each row, different superscript letters indicate significant difference (*p* < 0.05).

**Table 5 antioxidants-12-00700-t005:** DEGs enriched in KEGG pathways.

Gene Name	KEGG Pathway	Fold Change	*p* Value
*scd*	Biosynthesis of unsaturated fatty acids; PPAR signaling pathway	0.03	0.00
*elovl5*	Biosynthesis of unsaturated fatty acids	0.14	0.00
*elovl6*	Biosynthesis of unsaturated fatty acids	0.17	0.00
*fads*	Biosynthesis of unsaturated fatty acids; PPAR signaling pathway	0.21	0.00
*fabp7a*	PPAR signaling pathway	0.01	0.00
*cyp7a1*	PPAR signaling pathway	5.58	0.00
*angptl4*	PPAR signaling pathway	0.00	0.00
*fabp3*	PPAR signaling pathway	0.15	0.00
*pparαb*	PPAR signaling pathway	0.01	0.00
*acsl4a*	PPAR signaling pathway	0.27	0.01
*cpt2*	PPAR signaling pathway	0.10	0.04
*abca5*	ABC transporters	0.13	0.01
*abca1b*	ABC transporters	0.08	0.00
*acat2*	Terpenoid backbone biosynthesis; Two-component system	3.55	0.00
*cyp51*	Steroid biosynthesis	3.22	0.00
*aspg*	Glycerophospholipid metabolism	4.95	0.00
*pisd*	Glycerophospholipid metabolism	3.37	0.00
*chkb*	Glycerolipid metabolism; Glycerophospholipid metabolism	0.07	0.01
*gpat4*	Glycerolipid metabolism; Glycerophospholipid metabolism	0.08	0.00
*lypla2*	Glycerophospholipid metabolism	0.24	0.01
*pnpla2*	Glycerolipid metabolism	5.10	0.02
*fasn*	Fatty acid biosynthesis; Insulin signaling pathway	0.04	0.01
*cdk2*	p53 signaling pathway	0.22	0.00
*gadd45β*	p53 signaling pathway	28.78	0.03
*sod1*	Peroxisome; Amyotrophic lateral sclerosis (ALS)	2.15	0.04
*wars*	Aminoacyl-tRNA biosynthesis	0.09	0.01
*igfbp3*	p53 signaling pathway	24.97	0.00
*nansa*	Amino sugar and nucleotide sugar metabolism	0.17	0.02
*dmgdh*	Glycine, serine and threonine metabolism	2.17	0.01
*hk1*	Fructose and mannose metabolism; Amino sugar and nucleotide sugar metabolism; Starch and sucrose metabolism	0.03	0.00

## Data Availability

The raw sequencing data have been deposited at Sequence Read Archive (SRA) under the accession number (PRJNA929353).

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
