# Peer review of "Transcriptional Inhibition of AGPAT2 Induces Abnormal Lipid Metabolism and Oxidative Stress in the Liver of Nile Tilapia Oreochromis niloticus"

_antioxidants, 2023, doi:10.3390/antiox12030700_

Round 1
Reviewer 1 Report
Authors reported detailed data on the effects of knock down of agpat2, which loss of function was known to cause systemic lipodystrophy, on the liver of tilapia, an economically important farmed fish. Overall, studies are very important and well-designed. However, it appears that some interpretations of KEGG analyses and their relationship to observed phenotypes are not always clear. Letters in Figure 5 and 6 are too small and I cannot read them well. Please make them larger for easy reading. Also, please put spaces between each section.
General comments:
1) Can Authors provide information on agpat2 transcripts or protein in an ideal living environment and in an inhospitable environment, such as one that is enclosed and prone to causing liver damage? Information on other fish besides tilapia is also welcome. If Authors can provide adequate information, please add it in the revised manuscript.
Specific comments:
(Abstract)
1) “1-acylglycerol…..” in L13 should be“1-Acylglycerol…..”.
2) “The results suggest that the PPAR signaling 25 pathway…” in L25-26: It's difficult to understand just by reading the summary. Suddenly, PPAR appears.
(Introduction and other parts)
3) A sentence from L91 to L93: How did Authors introduce specific antisense RNA sequences into eggs through micropores? Did they mean electroporation?
4) Figure 1: Why did Authors try conventional RT-PCR not qPCR? Authors might have been able to find good primer sets to produce a genuine single band? Anyway, images of “a” and “b” look very different. Can Authors modify it?
5) A sentence in L338-L339: Please provide sequencing results. Were 1100-1200 bp bands agpat2 and 1000 bp bands not?
6) L400: “A4–C9, agpat2 knock-down” should be “A4–A9, agpat2 knock-down”.
7) Figure 4: Can Authors provide information on changes of some house-keep genes by agpat2-knock-down?
8) Table 5: Since Authors suggested oxidative stress by gpat2-knock-down, Keap1-Nrf2 signaling pathway should be changed. Can Authors provide any information on it? Please add comments on glutathione S-transferase pi (GSTP). If Keap1-Nrf2 signaling was not affected, how do Authors explain that SOD1 was upregulated?
9) A paragraph from L602: Enhancement of P53 signaling was remarkable. Did Authors find pyknosis in the liver, although Authors provide HE-staining image of the liver section only?
Author Response
Point 1: Letters in Figure 5 and 6 are too small and I cannot read them well. Please make them larger for easy reading. Also, please put spaces between each section.
Response 1: We have modified Figure 5 and Figure 6, and spaces are left between each section, thank you for the reminder.
Point 2: Can Authors provide information on agpat2 transcripts or protein in an ideal living environment and in an inhospitable environment, such as one that is enclosed and prone to causing liver damage? Information on other fish besides tilapia is also welcome. If Authors can provide adequate information, please add it in the revised manuscript.
Response 2: In sticklebacks, agpat2 transcription levels were found to increase in the pectoral muscle in response to cold acclimation, and we added relevant information in Line 651-654.
Point 3: “1-acylglycerol…..” in L13 should be“1-Acylglycerol…..”.
Response 3: We have changed this in Line 13, thank you for the reminder.
Point 4: “The results suggest that the PPAR signaling 25 pathway…” in L25-26: It's difficult to understand just by reading the summary. Suddenly, PPAR appears.
Response 4: We have made corresponding modifications in the abstract, thank you for your suggestion.
Point 5: A sentence from L91 to L93: How did Authors introduce specific antisense RNA sequences into eggs through micropores? Did they mean electroporation?
Response 5: Antisense RNA sequences enter through micropores of the fertilized eggs by osmotic pressure in the buffer solution.
Point 6: Figure 1: Why did Authors try conventional RT-PCR not qPCR? Authors might have been able to find good primer sets to produce a genuine single band? Anyway, images of “a” and “b” look very different. Can Authors modify it?
Response 6: According to the previous studies of Qiang et al [1] and Yan et al [2] in our laboratory, conventional RT-PCR has been a mature and reliable verification method, so we adopted conventional RT-PCR. We are sorry that the images of ‘a’ and ‘b’ cannot be modified due to the difference in exposure of the gel imaging system, but we can confirm that, as expected, fragments of 1100-1200bp were detected in the agpat2 knockdown group, fragments of about 1000bp were detected in the negative control group, and no corresponding fragments were detected in the control group.
Point 7: A sentence in L338-L339: Please provide sequencing results. Were 1100-1200 bp bands agpat2 and 1000 bp bands not?
Response 7: The 1000 bp bands were the blank expression vector amplification product (negative control group). We have added on Line 380-382, thank you for your suggestion.
Point 8: L400: “A4–C9, agpat2 knock-down” should be “A4–A9, agpat2 knock-down”.
Response 8: We have changed this in Line 446, thank you for the reminder.
Point 9: Figure 4: Can Authors provide information on changes of some house-keep genes by agpat2-knock-down?
Response 9: The house-keep genes ldha, nono, arhgdia and rpl11 were significantly down-regulated. We have changed this in Line 459-460, Thank you for your advice.
Point 10: Table 5: Since Authors suggested oxidative stress by gpat2-knock-down, Keap1-Nrf2 signaling pathway should be changed. Can Authors provide any information on it? Please add comments on glutathione S-transferase pi (GSTP). If Keap1-Nrf2 signaling was not affected, how do Authors explain that SOD1 was upregulated?
Response 10: The defense mechanism formed by PPAR signal plays an important role in liver anti-oxidative stress. Studies[3] have shown that activation of PPAR inhibits SOD inactivation after anoxia/reoxygenation injury in hepatocytes, reduces ROS production and accelerates its clearance, alleviates oxidative stress injury caused by anoxia/reoxygenation injury in rat hepatocytes, and protects liver function. Therefore, the up-regulation of sod1 may be caused by the PPAR signaling pathway, and the specific mechanism will be further studied.
Point 11: A paragraph from L602: Enhancement of P53 signaling was remarkable. Did Authors find pyknosis in the liver, although Authors provide HE-staining image of the liver section only?
Response 11: We have changed it at Line 675-678, thank you for the reminder.
- Qiang, J.; Cao, Z.M.; Zhu, H.J.; Tao, Y.F.; He, J.; Xu, P. Knock-down of amh transcription by antisense RNA reduces FSH and increases follicular atresia in female Oreochromis niloticus Gene 2022, 842.
- Yan, Y.; Tao Y.F.; Cao, Z.M.; Lu, S.Q.; Xu, P.; Qiang, J. The Effect of Knocked-Down Anti-Mullerian Hormone mRNA on Reproductive Characters of Male Nile Tilapia (Oreochromis niloticus) through Inhibition of the TGF-Beta Signaling Pathway Fishes, 2022, 7.
- Chen, K.; Li, Y.H.; Xu, S.Q.; Hu, S.H.; Zhang, L. Protective Effects of Peroxisome Proliferator-Activated Receptor-alpha Agonist, Wy14643, on Hypoxia/Reoxygenation Injury in Primary Rat Hepatocytes Ppar Research, 2012, 2012.

Reviewer 2 Report
Review
Paper title: Transcriptional inhibition of AGPAT2 induces abnormal lipid metabolism and oxidative stress in the liver of Nile tilapia Oreochromis niloticus
The authors studied the effects of transcriptional inhibition of 1-acylglycerol-3-phosphate O-acyltransferase on lipid metabolism and stress markers in the liver of Nile tilapia Oreochromis niloticus. The authors found remarkable changes in growth performance characteristics and lipid and insulin profiles. The authors conducted comparative transcriptome analyses to reveal the most important pathways in altering hepatic lipid metabolism and oxidative stress in this fish.
All these reasons explain the relevance of the paper by Tiantian Feng and co-authors submitted to "Antioxidants".
General scores.
The data presented by the authors are original and significant. The study is correctly designed and the authors used appropriate sampling methods. In general, statistical analyses are performed with good technical standards. The authors conducted careful work that may attract the attention of a wide range of specialists focused on the biology and cultured tilapia.
Recommendations.
The aim of this study should be highlighted more clearly.
Please, provide information about the study period and season.
L 167. Please, provide the reason for considering only male fish in this study.
Figure 3. Please, increase the font size
Figure 4. Please, increase the font size
Figure 5. Please, increase the font size
Figure 6. Please, increase the font size
The authors should update the discussion with the following papers:
Ayisi, C.L., Yamei, C., Zhao, J. L. Genes, transcription factors and enzymes involved in lipid metabolism in fin fish. Agri Gene 20187, 7-14.
Tapia, P.J., et al. Absence of AGPAT2 impairs brown adipogenesis, increases IFN stimulated gene expression and alters mitochondrial morphology. Metabolism 2020, 111, 154341.
Specific remarks.
L 21. Consider replacing “were significantly increased, while the malondialdehyde content was” with “significantly increased, while the malondialdehyde content ”
L 145. Consider replacing “The eggs placed” with “The were eggs placed”
L 150. Consider replacing “then 0.8 mL” with “then 0.8 mL of”
L 178. Consider replacing “with 200 mg/L MS-222 solution” with “with 200 mg/L of MS-222 solution”
L 200. Consider replacing “The various indexes” with “Various indices”
L 215. Consider replacing “mixture was” with “mixture were”
L 223. Consider replacing “was added” with “were added”
L 355. Consider replacing “small letters” with “superscript letters”
L 361. Consider replacing “small letters” with “superscript letters”
L 365. Consider replacing “was significantly decreased” with “significantly decreased”
L 366. Consider replacing “were significantly increased” with “significantly increased”
L 402. Consider replacing “determine effect” with “determine the effects”
L 417. Consider replacing “down-regulated DEGs in” with “down-regulated DEGs in the”
L 418. Consider replacing “with control group” with “with the control group”
L 464. Consider replacing “significant difference” with “significant differences”
L 468. Consider replacing “aquaculture industry” with “the aquaculture industry”
L 494. Consider replacing “LDL has the” with “LDL has an”
Author Response
Point 1: The aim of this study should be highlighted more clearly.
Response 1: We have made changes in Line 26-30, thank you for your reminding.
Point 2: Please, provide information about the study period and season.
Response 2: We start the study in mid-May, when tilapia start breeding, and we start sampling in September.
Point 3: L 167. Please, provide the reason for considering only male fish in this study.
Response 3: Because of the different metabolic mechanisms of male fish and endowed fish, male fish grow faster than female fish, it makes more sense to focus on male fish, and all-male fish farming has high economic benefits in China
Point 4: Figure 3. Please, increase the font size.
Response 4: We have increased the font size in Figure 3, thank you for the reminder.
Point 5: Figure 4. Please, increase the font size.
Response 5: We have increased the font size in Figure 4, thank you for the reminder.
Point 6: Figure 5. Please, increase the font size.
Response 6: We have increased the font size in Figure 5, thank you for the reminder.
Point 7: Figure 6. Please, increase the font size.
Response 7: We have increased the font size in Figure 6, thank you for the reminder.
Point 8: The authors should update the discussion with the following papers:
Ayisi, C.L., Yamei, C., Zhao, J. L. Genes, transcription factors and enzymes involved in lipid metabolism in fin fish. Agri Gene 20187, 7-14.
Tapia, P.J., et al. Absence of AGPAT2 impairs brown adipogenesis, increases IFN stimulated gene expression and alters mitochondrial morphology. Metabolism 2020, 111, 154341.
Response 8: We have updated the papers in the discussion, thank you for your suggestions.
Point 9: L 21. Consider replacing “were significantly increased, while the malondialdehyde content was” with “significantly increased, while the malondialdehyde content ”
Response 9: We've replaced it in Line 22, thank you for your suggestion.
Point 10: L 145. Consider replacing “The eggs placed” with “The were eggs placed”
Response 10: We've replaced it in Line 160, thank you for your suggestion.
Point 11: L 150. Consider replacing “then 0.8 mL” with “then 0.8 mL of”
Response 11: We've replaced it in Line 166, thank you for your suggestion.
Point 12: L 178. Consider replacing “with 200 mg/L MS-222 solution” with “with 200 mg/L of MS-222 solution”
Response 12: We've replaced it in Line 203, thank you for your suggestion.
Point 13: L 200. Consider replacing “The various indexes” with “Various indices”
Response 13: We've replaced it in Line 227, thank you for your suggestion.
Point 14: L 215. Consider replacing “mixture was” with “mixture were”
Response 14: We've replaced it in Line 246, thank you for your suggestion.
Point 15: L 223. Consider replacing “was added” with “were added”
Response 15: We've replaced it in Line 254, thank you for your suggestion.
Point 16: L 355. Consider replacing “small letters” with “superscript letters”
Response 16: We've replaced it in Line 398, thank you for your suggestion.
Point 17: L 361. Consider replacing “small letters” with “superscript letters”
Response 17: We've replaced it in Line 405, thank you for your suggestion.
Point 18: L 365. Consider replacing “was significantly decreased” with “significantly decreased”
Response 18: We've replaced it in Line 410, thank you for your suggestion.
Point 19: L 366. Consider replacing “were significantly increased” with “significantly increased”
Response 19: We've replaced it in Line 411, thank you for your suggestion.
Point 20: L 402. Consider replacing “determine effect” with “determine the effects”
Response 20: We've replaced it in Line 448, thank you for your suggestion.
Point 21: L 417. Consider replacing “down-regulated DEGs in” with “down-regulated DEGs in the”
Response 21: We've replaced it in Line 466, thank you for your suggestion.
Point 22: L 418. Consider replacing “with control group” with “with the control group”
Response 22: We've replaced it in Line 467, thank you for your suggestion.
Point 23: L 464. Consider replacing “significant difference” with “significant differences”
Response 23: We've replaced it in Line 514, thank you for your suggestion.
Point 24: L 468. Consider replacing “aquaculture industry” with “the aquaculture industry”
Response 24: We've replaced it in Line 519, thank you for your suggestion.
Point 25: L 494. Consider replacing “LDL has the” with “LDL has an”
Response 25: We've replaced it in Line 545, thank you for your suggestion.
